# Distal Interphalangeal Joint Involvement May Be Associated with Disease Activity and Affected Joint Distribution in Rheumatoid Arthritis

**DOI:** 10.3390/jcm11051405

**Published:** 2022-03-04

**Authors:** Takahiro Mizuuchi, Tetsuji Sawada, Susumu Nishiyama, Koichiro Tahara, Haeru Hayashi, Hiroaki Mori, Eri Kato, Mayu Tago, Toshihiro Matsui, Shigeto Tohma

**Affiliations:** 1Department of Rheumatology, Tokyo Medical University Hospital, 6-7-1 Nishi-Shinjuku, Shinjuku, Tokyo 160-0023, Japan; mizu03tk@yahoo.co.jp (T.M.); ta8ra@tokyo-med.ac.jp (K.T.); haeru-3@tokyo-med.ac.jp (H.H.); morihiroaki0124@yahoo.co.jp (H.M.); kimueri1127@yahoo.co.jp (E.K.); pi-kucho.m@view.ocn.ne.jp (M.T.); 2Rheumatic Disease Center, Kurashiki Medical Center, 250 Bakuro, Kurashiki 710-8522, Japan; newcity@live.jp; 3Department of Rheumatology, Clinical Research Center for Allergy and Rheumatology, National Hospital Organization Sagamihara National Hospital, 18-1 Sakuradai, Minami, Sagamihara 252-0392, Japan; ninja02matsui@gmail.com; 4Department of Rheumatology, National Hospital Organization, Tokyo National Hospital, 3-1-1 Takeoka, Kiyose 204-8585, Japan; touma.shigeto.jy@mail.hosp.go.jp

**Keywords:** rheumatoid arthritis, distal interphalangeal joint, disease activity

## Abstract

We investigated the relationship between distal interphalangeal (DIP) joint involvement and disease activity in 10,038 patients with adult-onset rheumatoid arthritis (RA). The affected joint distribution was investigated using the joint indices (JI) *x*, *y*, and *z*, corresponding to the upper and lower joints, and the predominance of large-joint involvement, respectively. DIP joint involvement (defined by the presence of tenderness and/or swelling in DIP joints) was present in 206 (2.1%) of 10,038 patients with RA. Patients with RA exhibiting DIP joint involvement were significantly younger, and more frequently women. DIP joint involvement was positively associated with Disease Activity Score-28 using C-reactive protein, and clinical variables related to high RA disease activity, including JIs *x* and *y*, and was negatively associated with JI *z*. JI *x* was significantly higher than JI *y* in RA patients with DIP joint involvement. An odds ratio analysis revealed that small-to-medium sized and upper-extremity joints ranked first, second, and fourth among the eight variables significantly associated with DIP joint involvement. The correlation coefficients revealed that small-sized and upper-extremity joints ranked first and second among the five significant variables. DIP joint involvement, albeit rare, is significantly associated with high RA disease activity with predominance of small-sized and upper-extremity joints.

## 1. Introduction

Rheumatoid arthritis (RA) is a systemic autoimmune disease characterized by persistent erosive synovitis that results in cartilage and bone destruction, leading to joint deformities and functional impairment [1]. Bone erosions in RA typically occur at the edge of the joint (marginal erosions) (i.e., the bare area), where the bone is not covered either by the cartilage or the synovial joint capsule. Rheumatoid synovitis usually begins in the small joints of the hands and feet before spreading to larger joints [2].

Research has suggested that the distal interphalangeal (DIP) joints are spared from the swelling and pain that typically affects the small joints of the hands in patients with RA [1,3]. Therefore, when a patient presents with DIP symptoms, such as tenderness and/or swelling at disease onset, some rheumatologists would argue that the differential diagnosis should place an emphasis on osteoarthritis (OA) and psoriatic arthritis (PsA) rather than on RA [1,4]. Primary hand OA is typically associated with involvement of the DIP joints (Heberden’s nodes), proximal interphalangeal (PIP) joints (Bouchard’s nodes), and the first carpometacarpal joints [5]. Although OA is a degenerative disease of cartilage and bone remodeling leading to bone spur formation, central subchondral erosions (gull-wing appearance on radiographs) can be observed in a subset of patients with hand OA (i.e., erosive OA) [6,7,8,9]. PsA also affects the DIP joints, although the radiographic pattern is characterized by a combination of erosive osteolytic changes and new peri-articular bone formation, which may give rise to the characteristic mouse-ear sign and pencil-in-cup deformities [10,11].

Jacob et al. previously stated the following [12]: “Authors of the conventional textbooks place different emphasis on the degree of DIP joint involvement in RA: Harris believes that DIP disease is not a regular feature of RA [13], while Williams and McCarty believe that DIP involvement occurs frequently in RA [14,15].” It should be noted that Fleming et al. examined the patterns of joint involvement in early RA, using factor analysis, and identified nine patterns of affected joint distribution, including a subgroup with DIP joint involvement [2]. Furthermore, Jacob et al. performed a radiographic analysis of DIP joint involvement in 62 patients with seropositive RA, who were then compared with 50 age- and sex-matched controls. The authors noted that the frequency of DIP joint erosion was significantly higher in patients with seropositive RA than in controls, and that the DIP erosions associated with RA did not occur in isolation but in combination with coexisting MCP or PIP joint disease. However, these DIP erosions were not related to disease duration [12].

Nishiyama’s joint indices (JIs; *x*, *y*, *z*) are novel measures for RA that consist of the following three components: *x* and *y* are indices of RA disease activity in the upper and lower extremities, respectively, while *z* is an index reflecting the predominance of large-joint involvement [16,17]. The present study aimed to examine the frequency of DIP joint involvement in patients with RA, and to determine its relationship with RA disease activity and affected joint distribution, based on data obtained from the National Database of Rheumatic Diseases in Japan (*NinJa*) [18].

## 2. Materials and Methods

### 2.1. Data Source

*NinJa* is a nationwide, multicenter, observational database established in 2002 and contains clinical data for patients with RA treated in Japan [18]. All registered patients were diagnosed with RA by their attending physicians, according to standard diagnostic criteria for this condition [19,20]. Clinical data of patients were collected annually at any time point during the indicated year. In the present study, we utilized data from patients with adult-onset RA (>18 years of age) registered in the *NinJa* in 2018 (*NinJa* 2018) whose affected joint counts, modified Health Assessment Questionnaire (mHAQ) results, Steinbrocker radiographic stage and functional class [21], and serum levels of rheumatoid factor and C-reactive protein (CRP) were collected (*n* = 10,038). Clinical parameters used for statistical analysis included age, sex, disease duration, radiographic stage, functional class, pain visual analog scale (VAS) score, tender joint counts (TJC) based on 68-joint counts, swollen joint counts (SJC) based on 66-joint counts [22], 28-joint Disease Activity Score based on C-reactive protein (DAS28-CRP) [23], global assessments of disease activity (Patient Global Assessment [PGA]), Physician’s Global Assessment of Disease Activity (PhGA), mHAQ, rheumatoid factor levels, and anti-cyclic citrullinated peptide (CCP) antibody levels. Pain VAS, PGA, and PhGA were assessed using a 10-cm scale.

The *NinJa* study protocol was reviewed and approved by the Ethics Committee of the National Hospital Organization (NHO) of Sagamihara Hospital (approval number: 2014031816), and by the ethics committee of each participating institution. All methods were performed in accordance with the relevant guidelines and regulations in Japan. Informed consent was obtained from all the participants included in the study.

### 2.2. Composite Disease Activity Index for RA

DAS28-CRP was calculated based on TJC and SJC (28-joint count), PGA results, and CRP levels [23]. RA disease activity was categorized according to DAS28-CRP scores as follows: remission (<2.3), low (≤2.7), moderate (≤4.1), or high (>4.1) [24].

### 2.3. DIP Involvement

The presence of DIP involvement was defined as the presence of tenderness and/or swelling in the DIP joints, except for the first interphalangeal joints. The number of DIP joints involved was thus calculated based on the sum of the second to fifth symptomatic DIP joints.

### 2.4. Joint Index

Nishiyama’s joint index was used to assess RA activity and affected joint distribution based on three components (*x*, *y*, *z*) [16]. Briefly, the joints were divided into the following four regions: upper/large (UL; shoulder, sternoclavicular, elbow, and wrist joints), upper/small (US; proximal interphalangeal (PIP) and metacarpophalangeal [MCP] joints), lower/large (LL; hip, knee, ankle, and tarsometatarsal joints), and lower/small (LS; metatarsophalangeal joints). The JI of each region was defined as the sum of tender and swollen joint counts divided by the number of evaluable joints within each region. The JI for the upper extremities (designated as *x*) and that for the lower extremities (designated as *y*) were defined as the summation of the JI of the UL region (JI[UL]) plus the JI of the US region (JI[US]) and that of the JI of the LL region (JI[LL]) plus the JI of the LS region (JI[LS]), respectively. The component *z* was defined as the JI of the large joints minus the JI of the small joints (JI[UL] + JI[LL] − JI[US] − JI[LS]), corresponding to the predominant involvement of large joints over small joints.

### 2.5. Statistical Analysis

Pearson’s chi-square tests were used to examine differences in clinical parameters for categorical variables. Adjusted standardized residuals (ASR) were used for multiple comparisons, whereby absolute values of ASR that were higher than 1.96 and 2.58 were considered to correspond to significance levels of *p* < 0.05 and *p* < 0.01, respectively. The relationships between two continuous parameters were evaluated using Pearson’s correlation coefficients. Statistical analysis was performed using SPSS version 26 (IBM Corp., Armonk, NY, USA) and JMP version 12.0.1 (SAS Institute Inc., Cary, NC, USA). All significance levels were set at *p* < 0.05 (two-sided).

## 3. Results

### 3.1. DIP Involvement in RA

Among the 10,038 included patients with RA with available joint counts, mHAQ results, and serum levels of CRP/rheumatoid factor, 206 (2.1%) presented with DIP joint involvement (i.e., tenderness and/or swelling in the second to fourth DIP joints). The distribution of affected DIP joint counts is shown in Figure 1 (median: 1, inter-quartile range [IQR]): 1–2). As for its laterality, the frequency of DIP joint involvement in the right hand and that in the left hand were 66.0% and 60.7%, respectively. The distribution of DIP joint involvement in each finger was similar between the right hand and the left hand, with the DIP joints of the index and middle fingers being the most frequently affected numerically, followed by the little finger, and the ring finger (Figure 2). As for the degree of activity of DIP synovitis, 112 of 206 (54.3%) patients with RA and DIP involvement had at least one swelling in their DIP joints.

### 3.2. Relationship between DIP Joint Involvement and RA Disease Activity

Clinical characteristics of patients with RA, with and without the presence or absence of DIP joint involvement, were shown in Table 1. Patients with RA exhibiting DIP joint involvement were significantly younger than those without such involvement. When the patients’ age distributions were categorized into 10-year intervals, the peak age of RA patients with DIP joint involvement was in the 60–69 years age group, whereas the peak age of those without DIP involvement was in the 70–79 years age group (Figure 3). DIP joint involvement was significantly more frequent among women than among men. The frequency of functional class 3–4 was significantly lower in patients with DIP joint involvement than in those without. However, disease duration, radiological stage III-IV, and rheumatoid factor positivity were not related to DIP joint involvement.

DAS28-CRP was significantly correlated with TJC, SJC, pain VAS, and PGA scores in addition to PhGA scores, with correlation coefficients of 0.66, 0.53, 0.67, 0.70, and 0.70, respectively (*p* < 0.01). JI *x* and JI *y* were significantly correlated with DAS28-CRP, with correlation coefficients of 0.80 and 0.40, respectively (*p* < 0.01). These clinical variables reflective of RA disease activity were significantly higher in patients with DIP involvement than in those without. Although serum CRP levels were lower in patients with DIP than in those without DIP, this difference was not statistically significant. RA disease activity was also analyzed based on DAS28-CRP. As shown in (Figure 4), DIP joint involvement was significantly associated with a higher frequency of high, moderate, and low disease activity (*p* < 0.01, *p* < 0.01, and *p* < 0.05, respectively), and with a lower frequency of remission status (*p* < 0.01).

We further investigated the relationship between DIP joint counts and RA disease activity markers. As shown in Table 2, the number of affected DIP joints was positively correlated with TJC, SJC, pain VAS, PGA, PhGA, DAS28-CRP, JI *x*, and JI *y* in patients with DIP joint involvement (Table 2). In addition, JI *x* was significantly higher than JI *y* in patients with RA exhibiting DIP joint involvement (*p* < 0.01).

### 3.3. Distribution of Affected Joints in Patients with RA Exhibiting DIP Joint Involvement

JI *z*, which represents the predominance of large-joint involvement, was significantly lower in patients with DIP involvement than in those without DIP involvement (Table 1). Furthermore, the number of affected DIP joints was negatively correlated with JI *z* in patients with DIP joint involvement (Table 2).

To determine the effect of DIP joint involvement on the distribution of affected joints in RA, we calculated the odds ratio of developing symptomatic joint involvement at the PIP, MCP, wrist, elbow, shoulder, hip, knee, ankle, and MTP joints, for patients with and without DIP joint involvement. As shown in Table 3, the PIP joint, MTP joint, MCP joint, wrist, ankle, shoulder, elbow, and knee were significantly associated with DIP joint involvement in descending order of odds ratios. Furthermore, symptomatic DIP joint counts in patients with DIP joint involvement were significantly correlated with those of the PIP joint, MCP joint, ankle, elbow, and hip in descending order of correlation coefficients (Table 4).

## 4. Discussion

In the present study, approximately 2% of patients with RA with relevant *NinJa* data exhibited DIP joint involvement. Furthermore, DIP joint involvement was positively associated with the female sex and RA disease activity markers, and negatively associated with age and JI *z*, an index for large-joint predominance.

The joints affected most frequently in RA are the small joints of the hands and feet, including the PIP, MCP, and MTP joints. The larger joints, such as those of the shoulders, elbows, wrists, hips, knees, and ankles, are also affected. However, previous studies have indicated that the DIP joints may be spared in patients with RA [1,3]. In our study, the prevalence of DIP joint involvement was 2% among RA patients enrolled in the *NinJa* 2018, and DIP joint involvement was associated with high RA disease activity. DAS28-CRP, a composite measure that reflects RA disease activity, was significantly higher in patients with DIP involvement than in those without. The variables that were significantly correlated with DAS28-CRP (including TJC, SJC, pain VAS, PGA, PhGA, JI *x*, and JI *y*) were all significantly associated with DIP joint involvement. In addition, DIP joint involvement was significantly associated with a high frequency of low, moderate, and high disease activity, and with a low frequency of remission status. Furthermore, DIP joint involvement increased the odds ratios of symptomatic involvement in other joints commonly affected in RA, indicating that DIP joint involvement does not occur alone, but in the presence of symptomatic non-DIP joints in the context of high RA disease activity. Therefore, it is possible that, for unknown reasons, the DIP joints are among the least affected joints in patients with RA, although they may become involved when RA disease activity is severe. In addition to its potential as a marker of disease activity, DIP joint involvement may characterize a new subtype of RA linked with high disease activity.

RA can be subdivided according to clinically relevant manifestations, such as age at RA onset (late-onset versus vs. early-onset) [25], positivity of autoantibodies (seropositive vs. seronegative) [26], and affected joint distribution (large-joint predominance vs. small-joint predominance) [27]. In the present study, DIP joint involvement was not related to either age at RA onset or autoantibody positivity. We used Nishiyama’s JIs to investigate the distribution of affected joints [16]. JI *x* and *y* reflect the degree of upper and lower joint involvement, respectively. Our findings showed that JI *x* was significantly higher than JI *y* in patients with DIP joint involvement, indicating that articular symptoms in these patients were more prominent in the upper extremities than in the lower extremities. Consistent with this finding, the correlation coefficients of the DIP joint with non-DIP joints were highest in combination with the PIP joint, followed by the MCP joint, ankle, elbow, and hip, where upper-extremity joints ranked first and second among the five statistically significant variables. A categorical analysis further revealed that the risk of developing symptomatic DIP joints was highest in patients with PIP joint involvement, followed by those with MTP, MCP, wrist, ankle, shoulder, elbow, and knee involvement. In this analysis, upper-extremity joints ranked first, third, and fourth among the eight statistically significant variables. Collectively, the data suggest that DIP joint involvement is linked to the involvement of upper-extremity rather than lower-extremity joints. In addition, DIP joint involvement was negatively associated with JI *z*, indicating that such involvement is linked to a predominance of small-joint involvement. This finding is compatible with the results of our odds ratio analyses, which revealed that small-to-medium sized joints (PIP, MCP, MTP, and wrist joints) ranked first through fourth among the eight variables significantly associated with DIP joint involvement. An analysis of correlation coefficients indicated that small-sized joints (PIP and MCP joints) ranked first and second among the five statistically significant variables. Further studies are required to determine whether RA characterized by a triad of DIP joint involvement, small-sized, and upper-extremity predominance, presents a new disease subtype associated with high disease activity.

The major strength of the present study is the large number of patients enrolled in the *NinJa*, which enabled us to characterize a rare subtype of RA with DIP joint involvement. Nonetheless, our study had several limitations. First, the *NinJa* database does not contain clinical data related to comorbidities, making it impossible to exclude the coexistence of RA with other rheumatic diseases, such as OA and PsA, in some patients. Previous studies have indeed demonstrated that OA can complicate RA, particularly in older adults [28,29,30]. In a cross-sectional analysis of 1988 patients with RA, Lechtenboehmer et al. demonstrated that radiographic DIP joint OA is present in up to 60% of patients, and that it is significantly associated with age, the female sex, and body mass index, but not with RA disease activity [30]. Regarding the location of affected DIP joints in OA, Rees et al. demonstrated that the DIP joint of the index finger is most frequently affected, while the ring finger is relatively spared in patients with hand OA [31]. The distribution pattern appeared to be similar to that of RA patients described here. However, it should be noted that the RA subgroup with DIP involvement described in the present study was distinct from that of patients with both RA and DIP joint OA as described by Lechtenboehmer et al., in that the former was associated with high disease activity. It should also be noted here that McAlindon et al. demonstrated that erosive hand OA was significantly associated with older age [9]. In contrast, RA patients with DIP involvement were significantly younger than those without involvement, in terms of numerical age (years) and the categorical percentage of the elderly. Although the impact of the statistical difference was not large, our findings suggest that RA with DIP involvement cannot be simply explained by incidental complications caused by OA. Further radiological studies, such as power Doppler sonography, would be useful to demonstrate the presence of active DIP synovitis. As for PsA, patients with psoriasis exhibiting musculoskeletal manifestations are rarely diagnosed with concomitant RA in daily clinical practice [32]. Recently, Chen et al. performed a retrospective analysis based on medical chart reviews and telephone interviews, demonstrating that the prevalence of unequivocal RA among patients with psoriatic disease was 0.23% [33]. Future studies should investigate the prevalence of comorbid RA and PsA in Japan.

Another limitation is the lack of data regarding the dominant hand. We showed that DIP joint involvement was numerically more frequent in the right hand in RA. However, the influence of left- or right-hand dominance on the laterality of DIP joint involvement cannot be analyzed using the *NinJa* database, since it does not contain the relevant data. Our study is also limited by the lack of radiographic analysis. Radiologically, OA is characterized by bony proliferation, including osteophyte formation (bone spurs), subchondral sclerosis, and cartilage degeneration, as manifested by narrowing of the joint space [5]. Erosive OA is characterized by a combination of central articular erosions (gull-wing appearance) and interphalangeal ankylosis in addition to bony proliferation features [6,7,8]. PsA is characterized by a combination of bony proliferation and bone resorption that leads to mouse-ear signs and pencil-in-cup deformities in the DIP joints [10,11]. Given that hand X-ray films were available in the *NinJa* database, it would have been helpful to confirm the absence of overlap with other rheumatic diseases. Finally, the racial homogeneity of patients with RA enrolled in the *NinJa* database represents a limitation of the present study. Therefore, the findings in the present study need to be confirmed by similar studies with different ethnic groups before generalization to patients with RA worldwide.

## 5. Conclusions

Our findings indicated that DIP joint involvement was significantly associated with high disease activity in patients with RA, particularly in patients with non-DIP involvement in the PIP joints, MCP joints, MTP joints, and wrists. Therefore, clinicians should remain aware of the potential for DIP joint involvement when evaluating disease activity in patients with RA. Furthermore, it remains essential to carefully discriminate RA from other rheumatic diseases that can affect the DIP joints, such as OA and PsA. Given the increasing number of late-onset RA due to the rapidly aging population [34], clinicians should also consider the potential for comorbid RA and OA overlap in older adults. Further studies are required to determine whether DIP joint involvement characterizes a new subtype of RA with high disease activity and predominance of small-sized and upper-extremity joints.

## Figures and Tables

**Figure 1 jcm-11-01405-f001:**
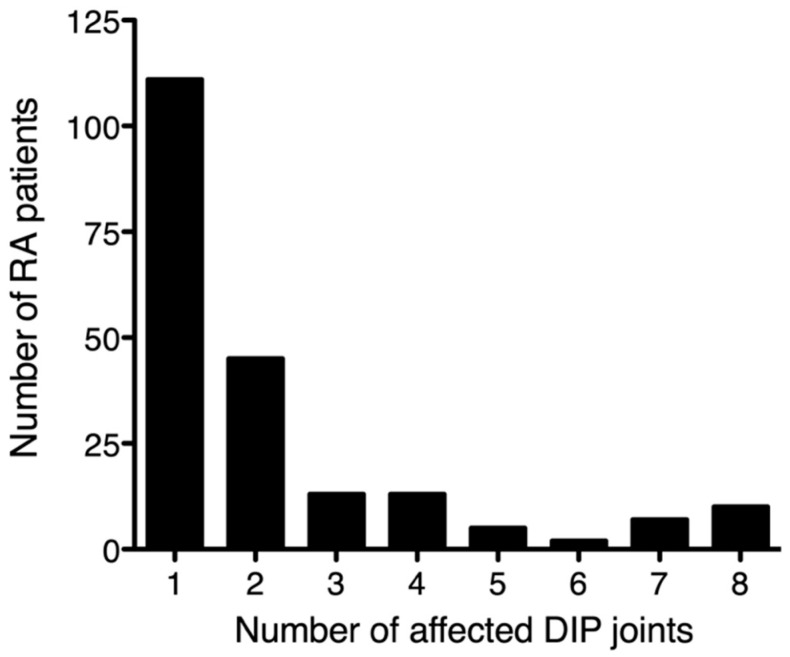
Distribution of symptomatic distal interphalangeal (DIP) joint involvement in 10,038 patients with rheumatoid arthritis (RA). The horizontal axis indicates the number of symptomatic DIP joints, as defined by the presence of tenderness and/or swelling at the second through fifth DIP joints. The vertical line indicates the number of patients with RA with each indicated number of symptomatic DIP joints.

**Figure 2 jcm-11-01405-f002:**
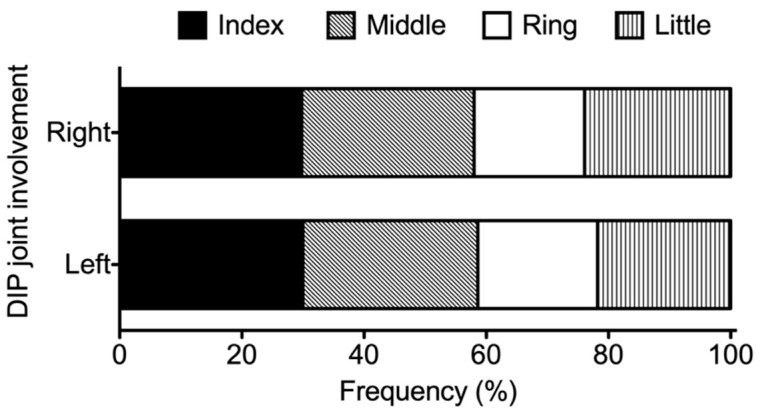
The distribution of distal interphalangeal (DIP) joint involvement in each finger in the right hand (upper) and the left hand (lower) in rheumatoid arthritis (RA) patients with symptomatic DIP joint involvement. Symptomatic DIP joint involvement was defined by the presence of tenderness and/or swelling from the second through fifth DIP joints. The frequency of DIP joint involvement of the index finger (second DIP joint), the middle finger (third DIP joint), the ring finger (fourth DIP joint), and the little finger (fifth DIP joint) were plotted horizontally as percentages.

**Figure 3 jcm-11-01405-f003:**
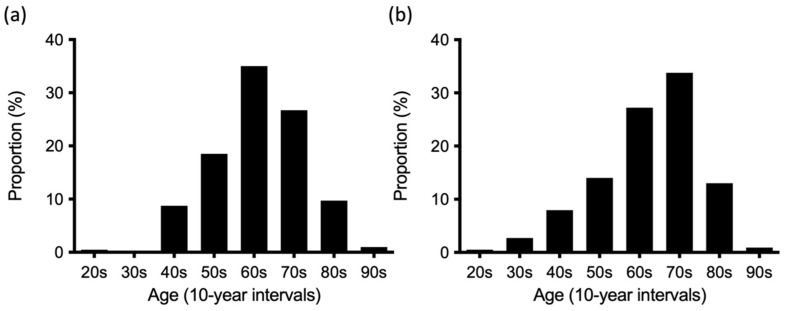
Age distribution of patients with rheumatoid arthritis (RA) with symptomatic distal interphalangeal (DIP) joint involvement (**a**) and those without it (**b**) in the *NinJa* database. Symptomatic DIP joint involvement was defined by the presence of tenderness and/or swelling from the second through fifth DIP joints. The horizontal axis indicates the distribution of RA patients’ age (10-year intervals). The vertical axis indicates the proportion (percentage) of RA patients in each age group.

**Figure 4 jcm-11-01405-f004:**
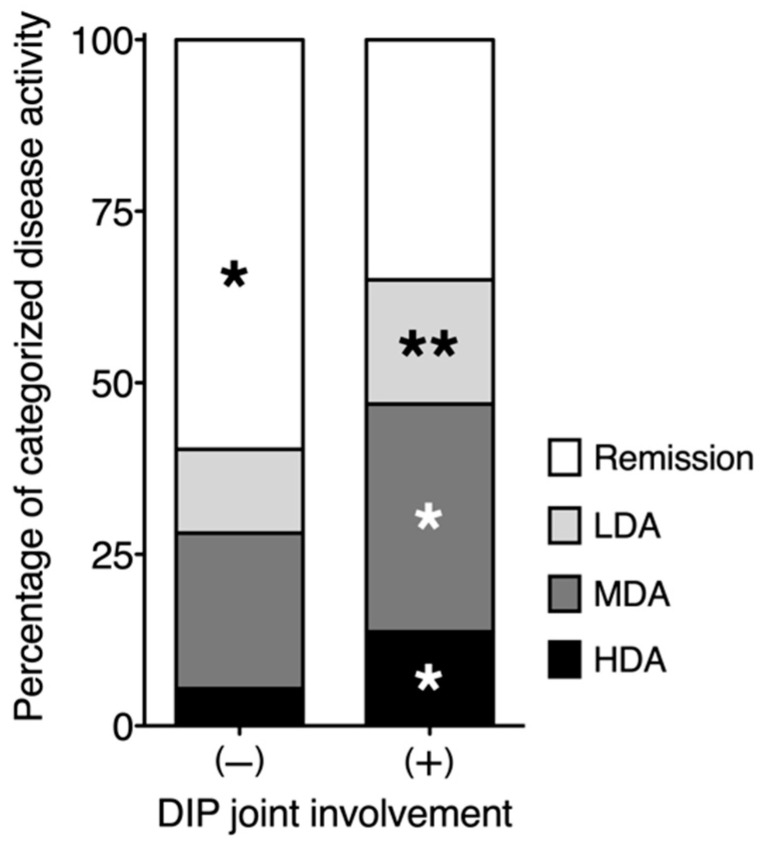
Distribution of disease activity for 10,038 patients with rheumatoid arthritis (RA) in the absence or presence of symptomatic distal interphalangeal (DIP) joint involvement. Symptomatic DIP joint involvement was defined by the presence of tenderness and/or swelling from the second through fifth DIP joints. The vertical axis shows the proportions of patients categorized as having remission (28-joint Disease Activity Score based on C-reactive protein [DAS28-CRP] < 2.3), low disease activity (LDA, 2.3 ≤ DAS28-CRP < 2.7), moderate disease activity (MDA, 2.7 < DAS28-CRP ≤ 4.1), and high disease activity (HDA, DAS28-CRP ≥ 4.1) in a stacked bar graph. * *p* < 0.01, ** *p* < 0.05.

**Table 1 jcm-11-01405-t001:** Clinical characteristics of patients with rheumatoid arthritis (RA) in the presence or absence of distal interphalangeal (DIP) joint involvement.

	DIP Involvement		
	Presence (*n* = 206)	Absence (*n* = 9832)	*p*-Value
Age, mean ± SD years	65.5 ± 11.3	66.8 ± 12.6	<0.05
Older adult population (≥65 years of age)	115 (55.8%)	6301 (64.1%)	<0.05
Female	182 (88.4%)	7842 (79.8%)	<0.01
Age at RA onset, mean ± SD years	52.9 ± 14.7	52.2 ± 14.8	NS
Disease duration, mean ± SD years	13.3 ± 10.4	13.9 ± 11.1	NS
Stage III-IV	79 (38.4%)	4310 (43.8%)	NS
Class 3–4	23 (11.2%)	1793 (18.3%)	<0.01
TJC, mean ± SD cm *	5.8 ± 8.3	1.8 ± 3.9	<0.001
SJC, mean ± SD cm *	4.7 ± 5.2	1.3 ± 2.7	<0.001
Pain VAS, mean ± SD cm *	3.0 ± 2.4	2.4 ± 2.3	<0.001
PGA, mean ± SD	3.1 ± 2.4	2.4 ± 2.2	<0.001
PhGA, mean ± SD	2.3 ± 1.5	1.5 ± 1.5	<0.001
mHAQ, mean ± SD	0.41 ± 0.60	0.38 ± 0.60	NS
DAS28-CRP, mean ± SD	2.9 ± 1.1	2.3 ± 1.0	<0.001
CRP mean ± SD mg/dL	0.48 ± 1.1	0.54 ± 1.2	NS
Positive rheumatoid factor	143 (69.4%)	7206 (73.3%)	NS
Positive anti-CCP Ab (*n* = 4835 **)	69/97 (71.1%)	3428/4738 (72.4%)	NS
Joint index *x*, mean ± SD	0.36 ± 0.39	0.17 ± 0.28	<0.001
*y*, mean ± SD	0.30 ± 0.41	0.12 ± 0.27	<0.001
*z*, mean ± SD	−0.04 ± 0.44	0.08 ± 0.30	<0.001

SD, standard deviation; TJC, tender joint counts; SJC, swollen joint counts; VAS, visual analog scale; PGA, Patient’s Global Assessment of Disease Activity; PhGA, Physician’s Global Assessment of Disease Activity; mHAQ: modified Health Assessment Questionnaire; DAS28-CRP: 28-joint Disease Activity Score based on C-reactive protein; Anti-CCP Ab: anti-cyclic citrullinated peptide antibody; NS: not significant. Percentages are included in parentheses. * Pain VAS, PGA, and PhGA were measured on a 10-cm scale. ** Anti-CCP Ab was measured in 4835 patients, and DIP joint involvement was present in 97 patients. The median titers of anti-CCP Ab were not significantly different between patients with RA with DIP involvement (median 48.5: 95% confidence interval [CI] 0.3–658.7) and those without it (median 56.4: 95% CI 0.5–1162.8).

**Table 2 jcm-11-01405-t002:** Correlation coefficients among the numbers of affected distal interphalangeal (DIP) joints and clinical variables in patients with rheumatoid arthritis (RA) exhibiting DIP joint involvement.

	DIP Count	TJC	SJC	Pain VAS	PGA	PhGA	mHAQ	DAS28-CRP	JI *x*	JI *y*	JI *z*
DIP count	1										
TJC	0.43 **	1									
SJC	0.30 **	−0.03	1								
Pain VAS	0.17 *	0.50 **	−0.02	1							
PGA	0.18 *	0.51 **	−0.02	0.93 **	1						
PhGA	0.19 **	0.58 **	0.21 **	0.54 **	0.53 **	1					
mHAQ	0.09	0.40 **	−0.03	0.41 **	0.46 **	0.21 **	1				
DAS28-CRP	0.26 **	0.73 **	0.21 **	0.71 **	0.74 **	0.63 **	0.43 **	1			
JI *x*	0.32 **	0.76 **	0.45 **	0.45 **	0.47 **	0.56 **	0.35 **	0.81 **	1		
JI *y*	0.23 **	0.67 **	0.38 **	0.25 **	0.26 **	0.54 **	0.31 **	0.48 **	0.60 **	1	
JI *z*	−0.16 *	−0.14	−0.39 **	0.06	0.10	−0.15 *	0.07	−0.03	−0.13	−0.23 **	1

TJC: tender joint count; SJC: swollen joint count; VAS: visual analog scale; PGA: Patient’s Global Assessment of Disease Activity; PhGA: Physician’s Global Assessment of Disease Activity; mHAQ: modified Health Assessment Questionnaire; DAS28-CRP: 28-joint Disease Activity Score based on C-reactive protein; JI: joint index. * *p* < 0.05, ** *p* < 0.01.

**Table 3 jcm-11-01405-t003:** Percentage of symptomatic involvement and odds ratios for non-distal interphalangeal (DIP) joints according to the presence or absence of DIP joint involvement.

	DIP Involvement			
	Presence (*n* = 206)	Absence (*n* = 9832)	Odds Ratio (95% CI)	*p*-Value
PIP joint	129 (62.6%)	1855 (18.9%)	7.2 (5.4–9.6)	<0.01
MCP joint	106 (51.5%)	2399 (24.4%)	3.3 (2.5–4.3)	<0.01
Wrist	90 (43.7%)	2598 (26.4%)	2.2 (1.6–2.9)	<0.01
Elbow	39 (18.9%)	1177 (12.0%)	1.7 (1.2–2.5)	<0.01
Shoulder	43 (20.9%)	1109 (11.3%)	2.1 (1.5–2.9)	<0.01
Hip	6 (2.9%)	165 (1.7%)	1.8 (0.8–4.0)	NS
Knee	45 (21.8%)	1454 (14.8%)	1.6 (1.2–2.3)	<0.01
Ankle	51 (24.8%)	1327 (13.5%)	2.1 (1.5–2.9)	<0.01
MTP joint	73 (35.4%)	898 (9.1%)	5.5 (4.1–7.3)	<0.01

CI, confidence interval; PIP, proximal interphalangeal; MCP, metacarpophalangeal; MTP, metatarsophalangeal; NS, not significant.

**Table 4 jcm-11-01405-t004:** Correlation coefficients among the numbers of affected distal interphalangeal (DIP) and non-DIP joints in patients with rheumatoid arthritis (RA) exhibiting DIP joint involvement.

	DIP	PIP	MCP	Wrist	Elbow	Shoulder	Hip	Knee	Ankle	MTP
DIP	1.00									
PIP	0.43 **	1.00								
MCP	0.33 **	0.55 **	1.00							
Wrist	0.11	0.18 *	0.32 **	1.00						
Elbow	0.18 **	0.29 **	0.45 **	0.36 **	1.00					
Shoulder	0.11	0.26 **	0.34 **	0.36 **	0.36 **	1.00				
Hip	0.17 *	0.11	0.07	0.06	0.04	0.12 ^#3^	1.00			
Knee	0.13 ^#1^	0.20 **	0.26 **	0.32 **	0.18 **	0.28 **	0.22 **	1.00		
Ankle	0.19 **	0.32 **	0.34 **	0.35 **	0.39 **	0.30 **	0.01	0.25 **	1.00	
MTP	0.13 ^#2^	0.39 **	0.36 **	0.25 **	0.26 **	0.23 **	−0.08	0.09	0.24 **	1.00

PIP, proximal interphalangeal; MCP, metacarpophalangeal; MTP, metatarsophalangeal. * *p* < 0.05. ** *p* < 0.01. ^#^ *p*-value below the level of statistical significance, albeit less than 0.1 (^#1^ *p* = 0.06, ^#2^ *p* = 0.07, ^#3^
*p* = 0.08).

## Data Availability

The datasets used in this study can be made available by the corresponding author upon reasonable request.

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
