# Peer review of "Distal Interphalangeal Joint Involvement May Be Associated with Disease Activity and Affected Joint Distribution in Rheumatoid Arthritis"

_jcm, 2022, doi:10.3390/jcm11051405_

Round 1
Reviewer 1 Report
I have reviewed the paper entitled: "Association of distal interphalangeal joint involvement with high disease activity and affected joint distribution in rheumatoid arthritis " (jcm-1598392) by Mizuuchi et al.
The authors investigate the relationship between distal interphalangeal (DIP) joint involvement and some features of patients with rheumatoid arthritis (RA) as well as the activity of the disease. To carry out the study, patient data included in a multicenter observational database of RA patients treated in Japan (National Database of Rheumatic Diseases in Japan, Ninja 2018 database) were used.
The most relevant results were: 1) the DIP involvement is rare because only 2.1% of patients had this feature (206 patients with DIP involvement out of the 10,038 with available data for joint counts, mHAQ results and serum level of C-reactive protein and rheumatoid factor). 2) Patients with and without DIP involvement showed some differences (patients with DIP were younger, with a higher ratio of females and more active disease). 3) Lastly, they found differences between patients with and without DIP in the distribution of the affected joints.
In my opinion, the paper is clear and of interest, with conclusions based on a large cohort of patients and statistical methods that seem adequate. The data could be valuable in the handling of RA patients. The study could help to characterize RA subtypes. The study has limitations, most of them cited by the authors in the Discussion. To mention another, the authors should state the ethnic homogeneity of the cohort. Generalizations to worldwide RA patients need confirmation of results in similar cohorts from different ethnic groups.
Author Response
We appreciate your valuable comment. Considering your suggestion, we have added the following sentences with track change in the last paragraph of the Discussion section.
Revised sentences: Finally, the racial homogeneity of patients with RA enrolled in the NinJa database represents a limitation of the present study. Therefore, the findings in the present study need to be confirmed by similar studies with different ethnic groups before generalization to patients with RA worldwide.
Reviewer 2 Report
Authors should state the study design, and the criteria that was used for the diagnosing RA in the NinJa data base.
the conclusion should just state the main findings without repeating the word CONCLUSION since the subheading is conclusion
Author Response
(1) Authors should state the study design, and the criteria that was used for the diagnosing RA in the NinJa data base.
We appreciate your valuable comment. We have now replaced the first sentence in the Database paragraph of the Methods section with a description of the study design and RA diagnostic criteria for the NinJa database.
Original sentence: NinJa is a nationwide, multicenter, observational database that contains clinical data for patients with RA treated in Japan (18).
Revised sentences: NinJa is a nationwide, multicenter, observational database that was established in 2002 and contains clinical data for patients with RA treated in Japan [18]. All registered patients were diagnosed with RA by their attending physicians according to standard diagnostic criteria for this condition [19, 20]. Clinical data of patients were collected annually at any time point during the indicated year.
(2) The conclusion should just state the main findings without repeating the word CONCLUSION since the subheading is conclusion.
We appreciate your valuable comment. As per your suggestion, we have deleted “In conclusion” from the first sentence of the Conclusion section.
Reviewer 3 Report
Manuscript ID: jcm-1598392
Reviewer’s comment:
The authors aimed to investigate the association of DIP joints involvement in RA patients. First, they demonstrated that DIP joint involvement was found in 2.1% in their RA cohort and that RA patients with DIP joint involvement were younger than those without DIP joint involvement and that they were more frequently women. Next, they also revealed that DIP joint involvement in RA was positively correlated with DAS28-CRP, clinical variables associated with HAD including JIs x and y, and was negatively correlated with JI z. In addition, they showed that JI x was significantly higher than JI y when DIP joint involvement was seen in RA. Furthermore, they demonstrated that small-to-medium sized and upper-extremity joints ranked first, second, and fourth among variables which were significantly associated with DIP involvement by odds ratio analysis. Finally, they revealed that small-sized and upper-extremity joints ranked first and second among some significant variables. They concluded that DIP joint involvement is significantly associated with high disease activity in RA especially as manifested by small-sized and upper-extremity joints.
The author's studies are very interesting; however, I have several major concerns.
Major concerns:
- The authors claimed the association of DIP joint involvement in RA patients especially with high disease activity; however, their findings did not have much conviction. It is said that the deformity or inflammation of DIP joints are occurred independent of RA pathology in rheumatologists. Indeed, they commented the meaning of DIP joint involvement in Discussion page 8, line 247-248, “DIP joint involvement does not occur alone, but in the presence of symptomatic non-DIP joints in the context of high RA disease activity”. As they described above, is the DIP joint involvement just a result including swan neck or button-hole deformity after deformity of MCP and PIP joints ? If so, significant meaning of DIP joint involvement in RA could not be a convincing argument.
- As related to above, I think that one of the hardest parts is to discriminate between OA and RA in DIP joint involvement. They discussed the age to explain the differences from the results in Table 1. Indeed, RA patients with DIP involvement were significantly younger than those without DIP involvement in their large cohort study; however, I think it should be considered with carefully because mean age were 65.5 and 66.8, respectively. Does the statistical significant differences have an impact for explaining clinical differences ?
- They showed that anti-CCP antibodies prevalence between two groups in Table 1 did not differ significantly. In many previous reports, RA disease activity is higher in patients with high titer anti-CCP Ab than in those with low titer anti-CCP Ab. Did ACPA-titer differ between two groups ?
In page 8, line 255; the sentence “DIP joint involvement may represent a novel prognostic factor for RA”, I think there is a gap in their hypothesis.
Author Response
(1) Is the DIP joint involvement just a result including swan neck or button-hole deformity after deformity of MCP and PIP joints? If so, significant meaning of DIP joint involvement in RA could not be a convincing argument.
We appreciate your insightful comments, raising the possibility that symptomatic DIP joint is simply caused by mechanical finger deformities such as swan neck or button-hole deformity. Since the NinJa database does not include data regarding the presence or absence of hand deformity in enrolled patients, it is difficult to describe the prevalence of hand deformity in the present study. However, 127 of 206 patients (61.7%) with RA having DIP involvement belonged to radiographic stage I/II (Table 1). In addition, 112 of 206 patients with RA complicated by DIP involvement (54.3%) had swollen DIP joints. Therefore, we consider that the symptomatic DIP involvement is not a simple sequel to hand deformities. As per your suggestions, we added the following sentences to 3.1. DIP involvement in RA paragraph of the results.
Revised sentence: As for the degree of activity of DIP synovitis, 112 of 206 (54.3%) patients with RA and DIP involvement had at least one swelling in their DIP joints.
(2) Indeed, RA patients with DIP involvement were significantly younger than those without DIP involvement in their large cohort study; however, I think it should be considered with carefully because mean age were 65.5 and 66.8, respectively. Does the statistically significant differences have an impact for explaining clinical differences?
We appreciate your thoughtful and insightful comments. We agree that the difference in age (years) and percentage of the elderly (>65 years old) between RA patients with DIP involvement and those without was small, albeit statistically significant (Table 1). According to your suggestion, we made the following changes to the sentences in the 3rd paragraph of the discussion.
Original sentences: In contrast, RA patients with DIP involvement were significantly younger than those without such involvement in the present study. Collectively, our findings suggest that RA with DIP involvement cannot be simply explained by incidental complications caused by OA.
Revised sentences: In contrast, RA patients with DIP involvement were significantly younger than those without involvement in terms of numerical age (years) and the categorical percentage of the elderly. Although the impact of the statistical difference was not large, our findings suggest that RA with DIP involvement cannot be simply explained by incidental complications caused by OA. Further radiological studies such as power Doppler sonography would be useful to demonstrate the presence of active DIP synovitis.
(3) Did ACPA-titer differ between two groups?
We appreciate your thoughtful comments. As per your suggestion, we added the following sentence to the footnote of Table 1.
Revised sentence: The median titers of anti-CCP Ab were not significantly different between patients with RA with DIP involvement (median 48.5: 95% confidence interval [CI] 0.3-658.7) and those without it (median 56.4: 95%CI 0.5-1162.8).
(4) In page 8, line 255; the sentence “DIP joint involvement may represent a novel prognostic factor for RA”, I think there is a gap in their hypothesis.
We appreciate your thoughtful comments. As per your suggestions, we have now deleted the sentence in the question.
Deleted sentences: Since the prevalence of rheumatoid factor or anti-CCP antibodies, which are predictors of joint destruction [25], did not significantly differ between patients with and without DIP involvement, DIP joint involvement may represent a novel prognostic factor for RA. Additional longitudinal radiographic studies are required to verify this hypothesis.
Round 2
Reviewer 3 Report
The authors carefully revised their manuscript. However, I am still skeptical of the association between DIP joint involvement and RA. They need to revise the title.
Author Response
1. The authors carefully revised their manuscript. However, I am still skeptical of the association between DIP joint involvement and RA. They need to revise the title.
We appreciate your valuable and insightful comments. Considering your suggestion, the title was replaced as shown below in order to implicate that the association of DIP joint involvement with the disease activity of RA can be still disputable and needs be elucidated in the future studies.
The original title: Association Of Distal Interphalangeal Joint Involvement with High Disease Activity and Affected Joint Distribution in Rheumatoid Arthritis
The revised title: Distal Interphalangeal Joint Involvement May Be Associated with Disease Activity and Affected Joint Distribution in Rheumatoid Arthritis